# Pathophysiology and Management of Fatigue in Neuromuscular Diseases

**DOI:** 10.3390/ijms24055005

**Published:** 2023-03-05

**Authors:** Francesca Torri, Piervito Lopriore, Vincenzo Montano, Gabriele Siciliano, Michelangelo Mancuso, Giulia Ricci

**Affiliations:** Department of Clinical and Experimental Medicine, Neurological Institute, University of Pisa, 56126 Pisa, Italy

**Keywords:** fatigue, neuromuscular disorders, metabolic myopathies, muscular dystrophies, mitochondrial diseases, spinal muscular atrophy

## Abstract

Fatigue is a major determinant of quality of life and motor function in patients affected by several neuromuscular diseases, each of them characterized by a peculiar physiopathology and the involvement of numerous interplaying factors. This narrative review aims to provide an overview on the pathophysiology of fatigue at a biochemical and molecular level with regard to muscular dystrophies, metabolic myopathies, and primary mitochondrial disorders with a focus on mitochondrial myopathies and spinal muscular atrophy, which, although fulfilling the definition of rare diseases, as a group represent a representative ensemble of neuromuscular disorders that the neurologist may encounter in clinical practice. The current use of clinical and instrumental tools for fatigue assessment, and their significance, is discussed. A summary of therapeutic approaches to address fatigue, encompassing pharmacological treatment and physical exercise, is also overviewed.

## 1. Introduction

Fatigue stands out as one of the most common symptoms in many chronic diseases such as cancer and multiple neurological disorders, including neuromuscular diseases. It has been extensively described and studied in diseases of the central nervous system (CNS) starting from multiple sclerosis and neurodegenerative diseases such as Parkinson’s disease and cerebrovascular disorders [1,2,3]; nowadays, it is also increasingly evaluated and investigated in disorders of the peripheral nervous system (PNS) such as chronic neuropathies, motoneuron diseases, and neuromuscular junction (NMJ) disorders and myopathies (Table 1). In this setting, fatigue becomes a further addition to other symptoms and carries a potentially disabling weight in terms of patients’ quality of life, including independence, adherence to physiotherapy, and working and social interactions. While more obvious and expected in NMJ disorders, some red flags such as the incapacity to recover from fatigue in a certain amount of time or by sleep, or fatigue being disproportionate to the task or also being present at rest [4,5,6] can be suggestive of a neuromuscular disorder, whether isolated or accompanied by other clinical features.

Kluger et al. [7] proposed a taxonomy that separates the concept of fatigue into two dimensions: (1) performance fatigability, meaning an objective measure of performance decrease over a period of time of exertion; (2) perceived fatigue, which includes all the subjective sensations that influence the homeostasis of the performer and is considered as a complex entity involving the central and peripheral nervous system along with metabolic alterations and, last but not least, psychological components. It is then evident that a significant amount of overlap with other terms and conditions, such as weakness and fatiguability, is present; hence, the distinction of these features and their assessment during patients’ examination are crucial. Nonetheless, to provide a comprehensive understanding of fatigue, several scenarios must be considered, with all of them being referrable under the same semantic category: fatigue can indeed manifest as a subjective feeling of tiredness variably associated with physical exertion, or as the need to interrupt a motor task, feeling unable to complete it, or applying maximum force to it.

Overall, the origin of those symptoms can be found in alterations of the CNS—a so-called “central fatigue”; of peripheral nerves, NMJ—peripheral fatigue; of muscles themselves (proper muscle fatigue); each component of this complex chain interplays with the others and multiple mechanisms can be involved as the underlying cause of fatigue.

Given its prevalence in neuromuscular disorders and its burden on normal life activities of patients, fatigue has been receiving growing attention with the development of more general or disease-specific scales and questionnaires that are also compared to physical performance in order to assess disease progression and efficacy of treatment. Determinants of fatigue in each kind of neuromuscular disorder are variable and range from macroscopic skeletal muscle or extramuscular changes to biochemical alterations in key points of cellular metabolism and homeostasis.

This narrative review aims to provide an overview on the pathophysiology of fatigue and its main determinants in some of the most common neuromuscular diseases such as muscular dystrophies, metabolic and mitochondrial myopathies, and Spinal Muscular Atrophy (SMA), and to underline the significance of fatigue as an outcome measure in clinical practice and trials; finally, present management strategies of fatigue as a symptom are discussed.

This review was not systematic and followed no specified protocol. We conducted our search on the PubMed database using the following keywords: “fatigue”, neuromuscular disease”, “metabolic myopathies”, “muscular dystrophies”, “mitochondrial diseases”, and “spinal muscular atrophy”. No standards or protocols guided the review. Except the iconic and historical articles, the most recent literature (last 10 years) was prioritized.

## 2. Pathophysiology of Fatigue

### 2.1. Muscular Dystrophies

Muscular dystrophies are hereditary degenerative disorders of skeletal muscle, sometimes accompanied by a multisystem involvement [8]. Fatigue is a frequent complaint in muscular dystrophies [9]. In a myopathic patient, fatigue can present acutely, during or at the end of effort, or as the impossibility to perform a motor task, mainly as a consequence of an energetic exhaustion or the absence of a minimum level of muscle strength required; or as a chronic, general condition, present also at rest and impacting on the patient’s capability to even begin the motor task and maintain it. Both central and peripheral fatigue are involved in muscular dystrophies: while central fatigue can be addressed to CNS involvement in the pathology and psychosocial factors, peripheral fatigue can be either due to abnormalities in the coupling of excitement–contraction in the muscle, the metabolic balance of substrates, and vasodilation phenomena in response to exercise by nitric oxide (NO) and cellular damage [10,11]. From a clinical point of view, in fact, acute fatigue can derive from the rupture of the sarcolemma after repetitive or eccentric exercise, which, if massive, can also degenerate in a chronic exhaustion that may last for days after exertion; on the other hand, several factors are involved in chronic fatigue, such as generally reduced mobility driving additional muscle wasting or increased energy requirements from contractures, deformities, and compensational strategies in walking, standing, and sitting. In particular diseases, such as Myotonic Dystrophy type 1, an avoidant personality is frequently accompanied by reduced motivation, depression, and perceived fatigue. In addition, respiratory involvement, very frequent in most muscular dystrophies, can impact through constant poor blood oxygenation and sleep disturbances, leading to daytime sleepiness and fatigue; this is also true for cardiological involvement, which is prominent in Dystrophinopathies, Myotonic Dystrophies, and some Limb Girdle Muscular Dystrophies (LGMDs).

At a biological level, the sarcolemma of striated muscle fibers is composed of the dystrophin–glycoprotein complex (DGC), which includes cytoskeletal proteins (dystrophin, syntrophins), the dystroglycan complex, and the sarcoglycan (SG) complex, providing mechanical linkage between the extracellular matrix and the cytoskeleton. The integrity of this complex structure is necessary to avoid sarcolemma damage and rupture during muscle contraction. LGMDs include four forms involving the four sarcoglycan glycoproteins, named sarcoglycanopathies (LGMD2D, 2E, 2C, and 2F), leading to an increased damage susceptibility of the sarcolemma. Also in dystrophinopathies, although at a different extent in Duchenne muscular dystrophy (DMD) compared to Becker muscular dystrophy (BMD), a similar damaging mechanism is present. The DGC also plays a role in molecular signaling, for example, through neuronal nitric oxide synthase (nNOS), which is anchored at the sarcolemma [12]. Nitric oxide (NO), formed by nNOS, is a molecule that regulates, among others, muscle contraction, blood flow and oxygenation, and cellular development. In physiological conditions, NO modulates sympathetic vasoconstriction. It has been hypothesized that loss of nNOS, which is normally anchored to the sarcolemma, in muscular dystrophies involving dystrophin and the DGC, may lower the amount of NO and its protective effect against ischemia that verifies during contraction, leading to higher reactive-oxygen species (ROS) and superoxides [13]. In early experiments on murine models, dystrophin loss was demonstrated to impair the vasoconstrictor response [14], with dystrophin-deficient mdx mice as well as nNOS null mice being unable to regulate muscle blood flow during exertion, leading to muscle focal necrosis. Fanin et al. [11] analyzed muscle biopsies from 32 patients with 7 forms of molecularly defined LGMD and 5 patients with DMD, providing evidence for a role of nNOS in affecting the disease phenotype. Moreover, sarcolemmal nNOS expression correlated with muscle fatigue and other clinical features, including dilated cardiomyopathy. Similar data come from studies on sarcoglycan-deficient muscle, with more pronounced NOs reduction in patients with complete SG complex deficiency (such as beta-sarcoglycanopathy) [15]. Recently, a model of reproduced contractile phenotypes from dystrophin-deficient myotubes using patient-derived iPSCs, with Electrical Field Stimulation (EFS)-based training programs, was studied by Uchimura et al. [16]. In their study, dystrophin does not appear as necessary to muscle fiber functional maturation, while it is required for cells contraction, implying functional deterioration rather than cellular damage, in contractile dysfunction. As one could speculate, inflammation could also be a player in fatigue generation; nonetheless, trials in DMD and sarcoglycanopathy patients with Deflazacort or Prednisone [17] did not find a correlation between the severe fatigue perceived by patients and the level of inflammation, while pain, lack of muscle force, and permanent weakness were related to fatigue. As for LGMDs and congenital myopathies, Caveolin-3 [18], Myotilin, and Titin [19] also seem to be involved in molecular intracellular signaling in vital processes such as cell regeneration and repair through nNOS.

Myotonic dystrophy type 1 (DM1) is the one of the most common muscular dystrophies in adult patients and is caused by an unstable expansion of CTG repeats in the *DMPK* gene; clinical features of DM1 include myotonia, mainly distal muscle weakness, and multisystemic involvement, including respiratory function, the heart, and the CNS. Fatigue and daily sleepiness are some of the main complaints from patients with DM1, with a multifactorial genesis of such symptoms, including obstructive sleep apneas, heart failure, and CNS involvement. In a study by Angelini et al. [9] involving muscle biopsies and brain MRI, the authors conclude that the muscle and brain are independently involved in DM1, with a central component of fatigue due to cortical atrophy and white matter lesions, and a peripheral component due to the atrophy of muscle fibers.

### 2.2. Metabolic Myopathies

Metabolic myopathies as a group, including muscle glycogenoses and lipid myopathies, share the impairment of the generation of energetic substrates in a sufficient manner to sustain exercise [20]. In these conditions, muscle weakness and atrophy can be present at variable extents, ranging from the sparing of muscular trophism to degenerative changes, while exercise intolerance, muscle pain, contractures—that need to be distinguished from cramps, as they are electrically silent at EMG—and rhabdomyolysis produced during or after exertion are common.

Glycogen storage disorders (GSD), or glycogenoses, are a vast family of inherited autosomal recessive disorders, each one characterized by reduced or altered enzymatic activity at a certain point of the glycolytic pathway; among them, acid alfa-glucosidase deficiency (Pompe disease, GSD II), Cori disease (GSD III), Andersen disease (GSD IV), myophosphorylase deficiency (McArdle disease, GSD V), phosphoglucomutase deficiency (GSD XIV), phosphorylase B kinase deficiency (GSD VIII), phosphofructokinase deficiency (Tarui disease, GSD VII), muscle phosphoglycerate mutase (GSD X) and kinase (GSD IX) deficiencies, muscle B-enolase deficiency (GSD XIII), and muscle lactate dehydrogenase deficiency (GSD XI) are the ones with a possible or prominent myopathic phenotype [21]. In this group, the most common forms in adults are Pompe disease, caused by biallelic mutations in the *GAA* gene encoding for a lysosomal enzyme responsible for total hydrolysis of glycogen to glucose, and McArdle’s disease, caused by pathogenic variants in the *PYGM* gene, encoding for the muscle-specific isoform of the glycogen phosphorylase enzyme, which plays a pivotal role in the first step of glycogenolysis, that is, the release of glucose-1-phosphate monomers in muscle fibers.

In these disorders, at a biochemical level, a variety of metabolic abnormalities are involved in reduced exercise tolerance and fatigue. Dosing lactate blood levels during and after effort (i.e., forearm ischemic test) show a fall of venous lactate levels during exercise in patients with McArdle disease and phosphofructokinase deficiency, due to the nearly complete metabolic block that characterizes these forms; in other muscle glycogenoses in which enzymatic activity is somewhat preserved at a certain level, lactate production is present, although not comparable with healthy subjects. Those processes lead to the absence of the normal fall in pH, leading to alkalosis, which is particularly true in McArdle’s disease, where one of the main affected pathways involves creatin kinase, resulting in higher AMP and ammonia production and reduced ATP creation. In McArdle’s disease, the impaired glycolytic process then might interfere with the membrane pump function and contribute to the abnormal fatigue and sarcolemmal damage [22]. Furthermore, also altered oxidative homeostasis has been proposed as a pathomechanism in muscle glycogenoses [23], thus expanding the consequences of these diseases also to aerobic metabolism, not only anaerobic. In fact, blocked glycogenolysis lowers the production of 4-carbon tricarboxylic acid (TCA) cycle intermediates (i.e., fumarate, malate, oxaloacetate), which slows the rate of generation of reducing agents such as NADH and FADH_2_, resulting in the impairment of energy production, and the reduced capacity of extracting oxygen from circulation results in an increased transport of oxygen by vessels. The various diseases differ based on the response to the presence of the extramuscular substrate: in the case of McArdle’s diseases, this is blatantly exposed by the “second wind” phenomenon, in which the patients are able to better sustain exertion after stopping and resuming exercise, which they performed with reduced capacity at the beginning, due to the capability of muscle cells to uptake glucose from the bloodstream and hepatic glycogen [24]. In Pompe disease, the enzymatic activity reduction results in impaired lysosomal degradation of glycogen, leading to the accumulation in tissues and altered glucose availability as an energetic substrate. At a molecular level, a variation in metabolome profile was demonstrated in KO mice models for Pompe disease, with a decrease in glycolysis and a metabolic shift from carbohydrate to lipids as an energy source [25]; similarly, the low glycolysis level in human primary myoblasts from Pompe disease patients and the progressive lysosomal glycogen accumulation in the tissue appear to fuel negative events such as altered autophagy and muscle proteostasis, and increments in oxidative stress that seem to be only partially relieved by Enzyme Replacement Therapy (ERT) [26]. Moreover, GSD II is often characterized by respiratory muscles involvement including both the diaphragm and accessory muscles, and sleep apneas are a common feature in late-onset patients; as seen before, poor blood oxygenation during the night hours can give rise to daytime sleepiness and fatigue in general. Moreover, in Pompe disease, the picture is further complicated by the presence of CNS involvement in infantile and juvenile forms, with structural changes at cortical, brainstem, and spinal cord levels, so an important central component of fatigue can be taken into account [27,28].

Lipid metabolic disorders can affect many tissues, among which the skeletal muscle and heart are frequently involved. Lipid myopathies (LMs), in particular, are rare diseases that, similarly to muscle glycogenoses, can present acutely or with a chronic course. Age at onset and clinical features are extremely variable, but in many cases, fatigue and exercise intolerance along with muscle weakness can be recognized [29]. Among them, Primary carnitine Deficiency (PCD) and Multiple Acyl-Coenzyme A Dehydrogenase Deficiency (MADD), in their adult, late-onset forms, are characterized by the presence of fatigue and exercise intolerance as main symptoms. PCD is an autosomal recessive disease caused by mutations in the *SLC22A5* gene, encoding for the Organic Cation/Carnitine Transporter 2 (OCTN2) protein. When manifesting in infancy or childhood, clinical features are severe and involve liver, skeletal, and cardiac muscle, with metabolic disturbances such as hyperammonemia and hypoketotic hypoglycemia, leading to hepatic encephalopathy. In adult-onset forms, which stand at the end of the clinical spectrum, the main clinical features include muscular weakness and fatigue [30]. Carnitine is necessary for the transport of long-chain fatty acids into the mitochondrial matrix, in order to proceed to fatty acid oxidation. Mutations impact on the transporter activity, with the residual level correlating with disease severity.

MADD, also known as Glutaric Acidemia Type II, is an autosomal recessive genetic disorder caused by mutations in the Electron Transfer Flavoprotein (*ETFA, ETFB*) or Electron Transfer Flavoprotein Dehydrogenase (*ETFDH*) genes, encoding for the α and β subunits of Electron Transfer Flavoprotein (ETF) and Electron Transfer Flavoprotein-Ubiquinone Oxidoreductase (ETF-QO) [31]. Mutations lead to impairment of the transfer of electrons from acyl-CoA dehydrogenases, compromising the β-oxidation process of fatty acids [31]. In a minority of cases, other genes involved in riboflavin transport are affected. MADD can present with a neonatal, severe form characterized by life-threatening metabolic imbalances, hypotonia, and cardiomyopathy, or an adult-onset form, frequently associated with mutations in the *ETFDH* gene, mainly expressed by exercise intolerance, myalgia, and fatigue. Both in PCD and MADD, the presence of fatigue is then ascribed to the impairment of the energetic production pathway involving fatty acids oxidation and alterations of cellular oxidative stress levels.

### 2.3. Mitochondrial Myopathies

Perceived fatigue and exercise intolerance are hallmark symptoms of primary mitochondrial diseases (PMDs) pointing out how mitochondrial dysfunction is a putative biological mechanism for fatigue.

PMDs are genetic metabolic disorders characterized by defects in oxidative phosphorylation with an estimated prevalence of 1:4300 [32]. Their pathophysiology is complex involving genetic mutations in both mitochondrial (mtDNA) and nuclear DNA with any inheritance pattern. PMDs are clinically heterogeneous, can occur at any age, and characteristically involve multiple systems, typically affecting organs that are highly dependent on aerobic metabolism [33]. Mitochondrial myopathies (PMMs) are a common manifestation of PMDs and are characterized by a predominant, but not exclusive, skeletal muscle involvement [34]. Extrinsic ocular muscles are commonly affected, manifesting as chronic progressive external ophthalmoplegia (PEO). Segmental muscles involvement may be isolated or associated with PEO [35]. Myopathies may be combined with additional “mitochondrial red flags” (such as sensorineural hearing loss, optic atrophy, peripheral neuropathy, stroke-like episodes, seizures, ataxia, failure to thrive, developmental delay/regression, cognitive decline, diabetes, short stature, cardiomyopathy, nephropathy, hepatopathy) or be part of a component of specific mitochondrial syndromes (i.e., Kearns–Sayre syndrome, MERRF) [32,33]. Clinical manifestations of mtDNA-related PMD depend largely on the relative proportion between normal and mutant mtDNA variants (threshold effect) [32]. The mutation load (heteroplasmy) at which muscle fibers’ oxidative capacity becomes impaired is closer to 65% in patients harboring mtDNA point mutations and 50% in those with large-scale mtDNA rearrangements (i.e., single deletion) [36].

Regardless of the phenotype, PMD patients experience exercise intolerance and fatigue in about 20% and 60–70% of cases, respectively [37,38], correlating with disease severity, with an increased risk of comorbid conditions such as depression, anxiety, and sleep disorders and lower psychosocial functioning and quality of life [35]. Mitochondrial dysfunction reduces muscle fibers’ oxygen extraction rates and their capacity to generate ATP, depletes phosphocreatine pools, and boosts lactate and free radical generation, increasing muscle dependence on anaerobic metabolism [35,38]. It has been shown that the increased levels of lactate in PMMs during rest and exercise is related to differences in lactate release/uptake rather than a global limitation in total lactate oxidation [38]. Moreover, the heavy reliance on anaerobic glycolysis of oxidative defective muscle fibers causes a fast depletion of glycogen storage, resulting in the recruitment of additional fibers for contraction, which, in turn, induces premature fatigue. Interestingly, muscle fuel metabolism preferences during exercise do not differ in PMM patients compared with healthy subjects [38]. At the whole-body level, in PMMs patients, the mtDNA mutation load inversely correlates with VO_2_max, which is the maximal rate of oxygen used for ATP production of the working muscle, and directly correlates with ventilator response rate during exercise; moreover, mitochondria have normal blood oxygen extraction rates at rest but are unable to increase the extraction level during exercise, resulting in an arterialization of muscular venous blood during fibers contraction. The limited oxygen extraction occurs along with an exaggerated cardiac output and a paradoxical local hyperemic response possibly mediated by the local release of ATP; this, in turn, overrides the vasoconstrictive effect of systemic norepinephrine released during exercise (“functional sympatholysis”) [35,38].

### 2.4. Spinal Muscular Atrophy

Spinal muscular atrophy (SMA) is an autosomal recessive disease resulting from homozygous mutations in the Survival Motorneuron 1 (*SMN1*) gene and the consequent production of a non-functioning SMN protein [39]. In affected individuals, disease severity is partially modulated by a variable expression of the SMN2 gene, depending on the number of copies of the gene, with higher SMN2 copies generally associated with milder phenotypes [40]. The SMN protein is present in all tissues [41], but the detrimental effect of the mutation is predominant in lower motor neurons, leading to muscle weakness, atrophy, and reduced motor function, across a phenotypical spectrum ranging from severe, life-threatening conditions in the perinatal era to late-onset forms. Along with weakness, fatigue is a common complaint in SMA patients. Apparently, motor neurons are not the only structures of the nervous system affected in SMA; in fact, neuromuscular junction (NMJ) abnormalities during development have been shown in SMA animal models [42] and in SMA patients with repetitive stimulation, supporting the connection between NMJ dysfunction and fatigue in this disease [43]. Moreover, skeletal muscle also shows abnormalities such as mitochondrial depletion and altered biogenesis in relation to SMN protein reduced expression [44]. A study by Montes et al. [45] focusing on the metabolic function of muscle during exercise with near-infrared spectroscopy in SMA patients reports decreased aerobic capacity to use oxygen for energetic metabolism as a failure in increasing deoxygenated hemoglobin despite the workload increase. In addition, evaluating the Six Minutes Walking Test (6MWT) and considering fatigue as a decrease in gait velocity from the first to the last minute, the authors found a 17% speed reduction in more than 2/3 of the patients, notwithstanding treatment with Nusinersen, while patients with neuromuscular diseases other than SMA did not show similar levels of fatigue. The authors conclude that SMA patients’ metabolism may be predominantly anaerobic, relying largely on glycolytic pathways, compared to healthy controls, and address mitochondria dysfunction and biogenesis downregulation—and the consequent NMJ and muscle impairment—to explain such abnormalities [46], which may be a downstream effect of SMN depletion.

Considering the tight relationship between fatigue and mitochondria, it is not surprising that abnormalities in mitochondrial biogenesis and function have been reported in animal and human models of SMA. Various studies have proved the impairment of mitochondrial respiration paralleled by reduced expression of nuclear and mitochondrial-encoded subunits of the OXPHOS chain [47], along with increased ROS and oxidative stress effects [48]; SMN1 and SMN2 transcription regulation is regulated in part by ROS, as demonstrated in experiments with paraquat (an oxidative-stress-inducing factor) that result in reduced SMN protein levels [49]. In addition, as the pivotal role of mitochondria in the activation of the intrinsic apoptosis’ pathway is well recognized [50], disturbances of molecular pathways involving Bcl-2 and p53 proteins have been demonstrated in SMA [51]. Finally, mitochondrial biogenesis appears to be impaired in SMA, if considering as a surrogate measure the amount of mtDNA and number of mitochondria, which has been reported in patient’s muscle tissue [52] with a correlation with disease severity and phenotype.

These findings are consistent with evidence coming from real-life data on treatment with Nusinersen, in which therapy reduces fatigue and increases walking distance at 6MWT [53]. Interestingly, recent data [43] show a failure in restoring NMJ function in adult patients treated with Nusinersen; the authors suggest as a possible explanation to this discrepancy that the intrathecally administered medication may not directly reach NMJ and peripheral tissues. Additionally, growing evidence from patients treated with “SMN-replacing” molecules show a residual degree of disease progression and only partially relieves it, suggesting that SMN depletion may not be the only causative mechanism in SMA to act against [54]. As newborn screening programs and treatment become increasingly available for SMA, addressing the pathophysiology and management of fatigue and the other ancillary symptoms that are not directly positively affected by SMN1-targeting medications becomes essential to optimize patients’ quality of life and overall health conditions.

A comprehensive overview on pathophysiological mechanisms involved in the generation of fatigue in the considered disorders is provided in Figure 1.

## 3. Fatigue Assessment and Utility as Outcome Measure

Assessment of both the perceived and performance-related dimensions of fatigue is crucial, also considering that it may not be obvious to recognize a strong association between the two, suggesting that many external factors (i.e., psychological, motivation, pain, and sleep disturbances) may be involved in determining fatigue as a whole; consequently, routine evaluations as well as clinical trials should include measuring tools capturing both. In this perspective, especially in adult patients with slowly progressive conditions, fatigue should be considered as an outcome measure, as it may vary at a different pace compared to other parameters such as strength, speed, or abnormalities at muscle MRI. In its subjective dimension, fatigue is commonly explored by self- or clinician-assessed questionnaires. Clinical and laboratory assessment tools for fatigue in neuromuscular diseases are summarized in Table 2.

### 3.1. Patients Reported Outcome Measures (PROMs)

The Fatigue Severity Scale (FSS), one of the most widely used assessment tools, was developed to measure subjective fatigue in patients with Multiple Sclerosis and Systemic Lupus Erythematosus; it has then been applied to multiple chronic conditions, including neurological and neuromuscular disorders. First published in 1989, the scale explores social, cognitive, and physical consequences of fatigue through 9 items, ranked from “strongly agree” to “strongly disagree” [55]. A study involving 225 adult Pompe disease patients demonstrated a high prevalence of the symptom with higher mean levels of fatigue in PD patients compared to healthy controls, without a clear correlation to sex, age, or disease duration, while increased fatigue was found in patients with respiratory involvement or using a wheelchair [56]; it is worth mentioning that in a study on ERT efficacy in LOPD patients, a reduction in the feeling of fatigue and augmented energy were among the first signs reported by patients [57], without a relationship with the motor improvement. The scale was also validated in SMA patients, where it did not show correlations with strength and motor endurance at 6MWT [58].

Another frequently employed scale in neuromuscular diseases and for clinical trials is the Checklist Inventory Scale (CIS), a 20-item questionnaire designed to assess fatigue severity, concentration, motivation, and physical activity, with high scores indicating high levels of fatigue. The scale was used to evaluate fatigue in patients in many neuromuscular disorders such as Oculopharyngeal Muscular Dystrophy (OPMD) [59], disclosing diffusely severe fatigue in the study cohort, and DM1 [60].

The Pediatric Quality of Life Includenventory (PedsQL) Multidimensional Fatigue Scale (MFS) is another commonly utilized tool exploring general fatigue, sleep-related fatigue, and cognitive fatigue in pediatric patients with chronic illnesses, including cancer, chronic kidney disease, and neuromuscular disorders. It was applied in DMD pediatric patients [61] and appears to correlate with depressive symptoms and sleep disturbances; it is also used in SMA [62]. The scale is particularly useful in children and adolescents with chronic neuromuscular diseases as it permits the identification of modifiable factors (such as sleep disorders and psychological issues) contributing to the symptom of fatigue.

The Multi-dimensional Fatigue Inventory (MFI) is another commonly used tool for the characterization of phenomenology and severity of fatigue, which has been validated for neuromuscular diseases such as ALS and SMA [63,64]. The scale investigates five dominions including general and physical fatigue, reduced activity, reduced motivation, and mental fatigue, with higher scores obtained indicating higher levels of fatigue. Kuo et al. [65] investigated fatigue in RYR1-related disorders with MFI and discovered a higher prevalence of fatigue in patients compared to the general population. Recently, Binz et al. described the use of MFI in a small group of adult SMA patients receiving Nusinersen treatment and found an overall reduction in fatigue with therapy over time [66].

### 3.2. Functional Outcome Measures

As for motor measures, one of the most frequently employed tests is the 6MWT, included also in many clinical trials for neuromuscular diseases such as DMD, SMA, and PMMs. Montes et al. demonstrated the peculiar changes in 6MWT in diverse neuromuscular diseases and its discriminative power between weakness, considered as a reduction in total distance walked compared to the prediction, and fatigue, computed as a d”crement in distance walked from the first to sixth minute. The authors includeded patients affected by dystrophinopathies, SMA, mitochondrial disorders, and other energy-depletion syndromes. The 6MWT appeared to optimally discriminate between conditions with predominant fatiguability more than weakness, such as SMA and mitochondrial diseases, further supporting the presence of varying and independent pathophysiologic mechanisms of fatigue in several neuromuscular conditions [67]. In DMD, the 6MWT has been historically employed as a major outcome measure in natural history studies and clinical trials, from steroids to AONs and small molecules [68,69], although more as a measure of weakness than fatiguability: along with the aforementioned work by Montes, a study from 2020 on 55 treated DMD boys failed to recognize a significant decrement in distance from the first minutes to the sixth, suggesting that 6MWT may not be a sensitive outcome measure to consider in trials for DMD with regard to fatigue [70]. Still, fatigue remains one of the main complaints of DMD patients; therefore, the need for motor scales able to capture it seems to remain still unmet. In Pompe disease, the 6MWT is used in natural history studies and to demonstrate efficacy of treatment with ERT, such as in the most recent trial for Avalglucosidase alfa (COMET study) [71], showing an overall improvement of distance and of motor endurance. The 6MWT is frequently used also in clinical practice and trials in PMM, as it also correlates with the perceived exercise intolerance, pain severity, and fatigue. 6MWT slope has also been proposed as a derived measure of fatigability in PMM [34]. More recently, based on the Endurance Shuttle Walk Test developed by Revill et al. [72] as a controlled paced walking test for chronic obstructive pulmonary disease, Bartels et al. elaborated a version suitable for assessment of fatiguability in non-ambulatory SMA patients, named Endurance Shuttle Box and Block test and 9HPT [73]. The Box and Block test measures upper limb fatiguability reproducing also daily life activities [74], while the 9HPT was validated in SMA 2 by Stam et al. [75].

### 3.3. Cycle ergometry, Ergoreflex Sensitivity, and Laboratory Measures

A useful tool in mitochondrial myopathies to assess exercise intolerance is cycle ergometry with measurements of VO_2_, VCO_2_, respiratory exchange ratio, heart rate, minute ventilation, rating of perceived exertion, and cardiac output. In PMMs, cardiopulmonary exercise testing shows a reduced VO_2_ peak and an early lactic acidosis [76]. Although the absolute values of blood lactate are not always higher than those of the normal population, the higher rate of lactate accumulation and bicarbonate buffering is responsible of abnormal values of respiratory exchange ratio [76]. PMM patients show increased cardiac output relative to VO_2_ level and ventilation. A linear correlation between peak VO_2_ and peak systemic arteriovenous O_2_ difference indicates that muscular oxidative capacity is limited by mitochondrial oxygen extraction rate [76]. Cardiopulmonary testing can also serve as a differential diagnosis tool, as different metabolic myopathies and mitochondrial diseases show different gas exchange and metabolite profiles in terms of V.V.O_2_ and lactic acid buffering by bicarbonate; conversely, disturbances of lipid metabolism do not show major abnormalities [76].

Ergoreflex sensitivity is another measure of fatigue in PMD patients. The ergoreflex is a neuromuscular reflex that regulates ventilation and the sympathetic response during exercise; its sensitivity can be quantified as the percentage of the ventilatory response to exercise maintained by post-handgrip regional circulatory occlusion compared to recovery without arm regional circulatory occlusion. Ergoreflex sensitivity is markedly enhanced in PMD patients compared with controls and correlates with several parameters of exercise performance and autonomic function [77,78]. PMD patients have a low anaerobic threshold, and lactic acidosis, even at rest, has been used as a pivotal biomarker for PMD diagnosis. However, several studies demonstrated its poor diagnostic sensitivity [32,79]. In any case, plasma lactate sampling during sub-maximal exercise and post maximal exercise represent a standardized method for exercise intolerance measurement in PMDs. During exercise, higher levels of lactate related to O_2_ utilization closely correlated with muscle oxidative defects [80]. 

## 4. Management and Treatment of Fatigue

Given the multifaceted nature of fatigue, its management may include several approaches working in parallel, from resolution of sleep disturbances to psychological intervention to pharmacological therapy directed to pathogenic factors and exercise therapy. Focusing on the main aforementioned determinants of fatigue, some experimental or already existing approaches are worth mentioning.

### 4.1. Pharmacological Treatment

#### 4.1.1. Muscular Dystrophies

In muscular dystrophies, muscular atrophy and motor deconditioning must be avoided at the maximum possible extent in order to preserve functional tissue and increase motor endurance. Supplements are commonly prescribed with variable results and controversial evidence [81]; for instance, carnitine is a well-known cellular metabolism supporter involved in fatty acid transport into mitochondria leading to ATP production and is frequently administered in myopathies. Nonetheless, clear scientific evidence for efficacy in improving exercise in a healthy population and muscle diseases is lacking [82]. Creatine is an anabolic compound that has been investigated in a series of chronic and acute conditions; its efficacy in improving muscle strength, the subjective feeling of energy, and activity in dystrophinopathies have been explored and confirmed by a series of studies [83,84,85,86]; Uchimura et al. [16], in their cellular model of contractile DMD myotubes, also demonstrated a partial rescue in contraction decline by administration of creatine. Different results come from studies on Myotonic Dystrophy type 1 and 2 that failed to demonstrate an improvement in physical function. On a more speculative side, based on the interaction between the cyclic Guanosine monophosphate (cGMP) pathway regulated by NO and nucleotide phosphodiesterase PDE5, a trial has been conducted with sildenafil, a PDE5 inhibitor, to improve cardiomyopathy, unfortunately without success [87]. The central role of NO in maintaining microcirculation and oxidative stress balance and its misregulation shared by dystrophinopathies, LGMDs, and others, nonetheless, underline its promising role as a modifiable factor to target.

#### 4.1.2. Metabolic Myopathies

Pompe Disease has witnessed the availability of an effective treatment through Enzyme Replacement Therapy (ERT) since the first approval by the FDA in 2006; the efficacy of ERT on strength and fatigue has been explored in several studies [88]; conversely, ERT shows a moderate effect in stabilizing ventilatory function [88,89].

In McArdle’s disease, given the substrate dependence for motor performance, one of the most common approaches involves dietary intervention with carbohydrate supplementation; nevertheless, a systematic review from 2014 did not succeed in identifying a clearly effective treatment in spite of low-quality evidence on the beneficial effect with supplements such as creatine, oral sucrose, ramipril, and a carbohydrate-enriched diet [90].

In PCD, both infantile and adult forms are treated by L-carnitine intravenous or oral supplementation. L-carnitine may also act through an antioxidant action in lowering fatigue and improving muscle strength [91]. Avoiding fasting and frequent meals also reduce the risk of hypoglycemia and metabolic imbalance that are frequent in this disorder. Several supplements have been anecdotally tried in lipid myopathies including PCD, MADD, and Neutral Lipid Storage Disorders (NLSD). MADD shows a certain degree of response to riboflavin [92]. Other compounds, such as CoQ10 and fenofibrates, have been reportedly tried in single patients with positive results [93,94,95].

#### 4.1.3. Mitochondrial Myopathies

In PMDs, treatment has been mostly focused on symptomatic management and supportive measures, such as treatment of catabolic stress and infections, exercise, and the use of a combination of vitamins and supplements (often referred to as “mito-cocktails”) [32]. A randomized clinical trial in 30 patients with PMDs showed a minor effect of Coenzyme Q10 on cycle exercise aerobic capacity and post-exercise blood lactate, even though it did not affect other variables such as strength or resting blood lactate [96]. Many molecules and compounds are under study in PMMs targeting, among others, fatigue and exercise tolerance: Elamipretide, a tetrapeptide that associates with cardiolipin, maintaining mitochondrial morphology, metabolic activity, and acting as an antioxidant, was assessed in a phase III, randomized, double-blind, placebo-controlled clinical trial (MMPOWER) for PMM patients. A course of 24 weeks of elamipretide administration did not meet the primary endpoints (6MWT and patient-reported fatigue score). However, a post hoc subgrouping analysis revealed a treatment effect based on the 6MWT in the nuclear DNA mutation subgroup. A new clinical trial is currently running to further investigate elamipretide in this subgroup of patients [97]. Furthermore, elamipretide was also tested in a phase II, randomized, double-blind, placebo-controlled crossover study with an open-label follow-up in patients with Barth syndrome (TAZPOWER). Barth syndrome is a PMD characterized by dilated cardiomyopathy, skeletal myopathy, neutropenia, and short stature. During the first 12 weeks, the study failed to meet its primary endpoints (6MWT and patient-reported fatigue scores improving), which were significantly improved at the open-label follow-up (36 weeks). Moreover, echocardiographic assessments indicated an improvement in cardiac function at the open-label follow-up [98]. Omaveloxolone, a semi-synthetic triterpenoid that potentiates NRF2 action on mitochondrial biogenesis, was assessed in a phase II, randomized, double-blind, placebo-controlled study (MOTOR trial) for PMM patients. Overall, 12 weeks of treatment with omaveloxolone were well tolerated. Although no significant differences in primary and secondary outcomes were observed (peak cycling exercise workload and 6MWT), treatment reduced heart rate and lactate levels during submaximal exercise, indicating improved mitochondrial bioenergetics and submaximal exercise tolerance [99,100]. NAD^+^ (nicotinamide adenine dinucleotide) is a cofactor for SIRT1, which activates mitochondrial biogenesis via PGC-1α. NAD^+^/NADH ratio control is also essential for intramitochondrial metabolic homeostasis. NAD^+^ deficiency was documented in PMM patients, and supplementation with NAD^+^ precursors had been shown to increase mitochondrial biogenesis and ameliorate the mitochondrial myopathy phenotype in animal models. Recently, an open-label study of niacin (nicotinic acid) in PMM patients was conducted. Overall, muscle strength and mitochondrial biogenesis increased in all subjects, and blood and muscle NAD^+^ reached control levels [101].

#### 4.1.4. Spinal Muscular Atrophy

Similarly to Pompe Disease, the natural history of SMA has been radically changed by the advent of SMN-replacing therapies, Nusinersen and Risdiplam, and, more recently, gene therapy [102]. Olexosime, a novel candidate treatment for SMA, was the first proposed to act on mitochondrial membrane permeability and proved effective on improving cellular survival. Unfortunately, after promising results in the phase II clinical trial showing positive effects on motor function compared to the placebo, the phase III trial was stopped as a consequence of unsatisfying long-term efficacy [103]. Nonetheless, given the multifactorial pathogenesis of weakness and decreased motor endurance in SMA, including mitochondrial dysfunction, this experience provides hints on the importance of working on molecules targeting also other factors than SMN expression, in order to provide patients with a comprehensive therapeutic strategy.

### 4.2. Physical Therapy

Finally, physical therapy and exercise remain fundamental components of global care of patients with neuromuscular disease, providing well-established benefits also on fatigue; obvious reasons for that are the preservation of muscular mass and maintenance of a certain level of activity along with prevention or treatment of joint contractures and altered posture. On a biochemical side, in a recent review, Grassi et al. [104] discussed evidence on the efficacy of moderate—also at home—training in metabolic and mitochondrial myopathies, which seem to exert a positive effect on exercise endurance and blood O_2_ peripheral extraction from muscle tissue estimated by near-infrared spectroscopy, although not on a global level of activity.

Several studies of animal models and human patients affected by PMMs demonstrated a beneficial effect of endurance exercise. In particular, training increases muscular mitochondrial content, stimulating biogenesis via PGC-1α activation and other mechanisms, improving antioxidant and OXPHOS enzyme activity, maximal oxygen uptake, and muscle strength, and decreasing rest and post-exercise blood lactate level [105,106]. Therefore, aerobic training could efficiently improve mitochondrial oxidative activity in muscles and quality of life of patients with PMDs.

## 5. Conclusions

Fatigue and exercise tolerance are core, although frequently overlooked, determinants of quality of life for patients affected by neuromuscular diseases. Their prevalence and significance, moreover, skyrocket in consideration of improved life expectancy and independence of patients. As growing knowledge on the pathophysiology of the different conditions permits the better understanding of the various components of central and peripheral fatigue, at a physiological and molecular level, an increasing number of therapeutic strategies are being approached with multiple targets, in parallel with more “traditional” or directed to primarily pathogenic factors (as in Pompe Disease, mitochondrial myopathies and SMA). Overall, this prompts the need to carefully assess both subjective fatigue and motor endurance in clinical practice, to better characterize diseases’ natural history, and in clinical trials as a useful outcome measure.

## Figures and Tables

**Figure 1 ijms-24-05005-f001:**
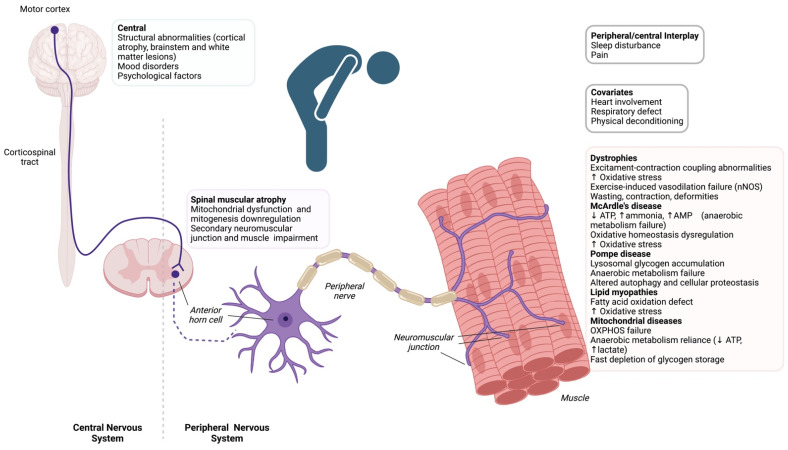
Overview on pathophysiological mechanisms involved in fatigue and exercise intolerance in neuromuscular disorders.

**Table 1 ijms-24-05005-t001:** Neuromuscular diseases associated with fatigue.

Neuromuscular Diseases Associated with Fatigue
Neuromuscular junction diseases (Myasthenia Gravis, Congenital Myasthenic Syndromes)
Muscle Channelopathies
Charcot–Marie–Tooth disease (CMT)
Guillain–Barre’ syndrome (GBS)
Infectious diseases (poliomyelitis, Lyme disease, viral infections)
Chronic inflammatory demyelinating polyneuropathy (CIDP)
Muscular Dystrophies
Metabolic (glycogenoses, lipid storage myopathies)
Primary mitochondrial diseases
Inflammatory Myopathies
Motor neuron disease (Amyotrophic Lateral Sclerosis—ALS, Spinal Muscular Atrophy—SMA, Spino-bulbar Muscular Atrophy—SBMA)

**Table 2 ijms-24-05005-t002:** Overview of fatigue assessment tools in different neuromuscular diseases.

Tool	Specific/Validated for
Patients reported outcome measures (PROMs)	
Fatigue Severity Scale (FSS)	Pompe Disease, SMA
Checklist Inventory Scale (CIS)	OPMD, DM1
Pediatric Quality of Life Inventory (PedsQL)	DMD, SMA
Multi-dimensional Fatigue Inventory (MFI)	SMA
Functional outcome measures	
6 Minutes Walking Test (6MWT)	
Box and Block test	SMA
9HPT	SMA
Cycle ergometry VO_2_, VCO_2_, respiratory exchange ratio, heart rate, minute ventilation, rating of perceived exertion cardiac output, lactate level and bicarbonate buffering	
Ergoreflex sensitivity	PMDs
Plasma lactate during and post sub-maximal exercise	PMDs

## Data Availability

Not applicable.

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
