# Peer review of "Pathophysiology and Management of Fatigue in Neuromuscular Diseases"

_ijms, 2023, doi:10.3390/ijms24055005_

Round 1

Reviewer 1 Report

In this manuscript, the authors review several aspects related with fatigue, including physiopathology, muscle glycogenosis and lipid myopathies, mitochondrial defects among others. English is good, and the manuscript is well written. I have some minor points:

line 39: decrease instead of "dicrease"

line 100: sarcoglycan instead of "sacroglycan"

line 118: Moreover instead of "moreover"

line 155: glycolytic instead of "glycolitic"

Line 185: FADH2 instead of FADH (FAD accepts two hydrogen atoms)

Line 220: of instead of "fo"

Line 243: an instead of "and"

Line 290: is instead of "in"

Line 350: evaluations instead of "evalutations"

Line 477: delete “and”

Line 553: (nicotinic acid) instead of (nico-tinic acid)

The + sign in NAD is superscript: NAD+

Author Response

Dear reviewer,
many thanks for your suggestion. We performed the corrections as suggested.

Reviewer 2 Report

Since this is a review paper, I would expect some explanation about the analysis of the literature. For example, if you look for articles about fatigue in different journals/editorials/repositories, which are the hot topics in the field and , a classification of papers organized according to data, experiments or metrics. 

I would also recommend a more schematic organization, including subsections, in section 3 and 4, and some tables to summarize the different topics covered in the paper (a kind of taxonomy of the different tasks).

Author Response

Dear reviewer,
many thanks for your suggestion. We performed the following corrections:
- added a method part in the introduction paragraph explaining the literature analysis of our narrative review
- added a table (Table 2) in the “Fatigue assessment and utility as outcome measure” paragraph summarizing the topic
- included subsections for paragraph 3 and 4
